# Legume-nodulating rhizobia are widespread in soils and plants across the island of Oʻahu, Hawaiʻi

**Jonathan N. A. Abe, Ishwora Dhungana, Nhu H. Nguyen**◉*

Department of Tropical Plant and Soil Sciences, University of Hawaiʻi at Mānoa, Honolulu, HI, United States of America

* nhu.nguyen@hawaii.edu

## Abstract

Legumes and their interaction with rhizobia represent one of the most well-characterized symbioses that are widespread across both natural and agricultural environments. However, larger distribution patterns and host associations on isolated Pacific islands with many native and introduced hosts have not been well-documented. Here, we used molecular and culturing techniques to characterize rhizobia from soils and 24 native and introduced legume species on the island of Oʻahu, Hawaiʻi. We chose two of these isolates to inoculate an endemic legume tree, *Erythina sandwicensis* to measure nodulation potentials and host benefits. We found that all rhizobia genera can be found in the soil, where only *Cupriavidus* was found at all sites, although at lower abundance relative to other more common genera such as *Rhizobium* (and close relatives), *Bradyzhizobium*, and *Devosia*. *Bradyrhizobium* was the most common nodulator of legumes, where the strain *Bradyrhizobium* sp. strain JA1 is a generalist capable of forming nodules on nine different host species, including two native species. In greenhouse nursery inoculations, the two different *Bradyrhizobium* strains successfully nodulate the endemic *E. sandwicensis;* both strains equally and significantly increased seedling biomass in nursery inoculations. Overall, this work provides a molecular-based framework in which to study potential native and introduced rhizobia on one of the most isolated archipelagos on the planet.

## Introduction

The Hawaiian Islands due to their isolation have developed a biogeographic landscape that is prone to invasion of introduced organisms, including soil microbes that range from pathogenic to mutualistic lifestyles. Although most microbial introductions were not recorded, the Nitrogen Fixation for Tropical Agricultural Legumes (NifTAL) Project, which started in Hawaiʻi in 1975, introduced a large number of non-native rhizobia to the islands [1]. The project expanded research on legumes, rhizobia, and inoculation methods that would increase the biological nitrogen fixation efficiency of legumes. At least some of the 1774 non-native rhizobia strains

**Data Availability Statement:** Sequence data is available under GenBank accession numbers MT703690-MT703747, OM980333-OM980337, and Sequence Read Archive BioProjects PRJNA551045, PRJNA661121, PRJNA661131,

PRJNA995197, PRJNA996221, PRJNA996231, PRJNA996312, and PRJNA996313; R analysis scripts are available on the GitHub repository for this project https://github.com/nnguyenlab/rhizobia.

**Funding:** This work was supported by the Undergraduate Research Opportunities Program (UROP) at UH M?noa, project 11930-JABE to JNAA, and the USDA National Institute of Food and Agriculture (NIFA), Hatch project 8042H to NHN, managed by the College of Tropical Agriculture and Human Resources. The funders had no role in study design, data collection and analysis, decision to publish, or preparation of the manuscript.

**Competing interests:** The authors have declared that no competing interests exist.

that came from this project [2–4] were used in experiments in Hawaiian soils to produce the inoculant technology that would eventually be spread across the tropical world [1, 5, 6].

Currently, the occurrence and distribution of rhizobia, whether introduced or not, and to which hosts they associate is not known. Hawaiʻi has 16 endemic legume species, 7 indigenous species, and at least 174 recorded introduced species [7]. Some introduced species, whether they arrived on purpose or by accident, are found on many, if not all, of the major Hawaiian Islands. These may be in the form of trees such as *Leucaena leucocephala* and *Prosopis pallida* that prefer open dryland habitats, or as many herbaceous weeds in the genera *Chamaecrista*, *Crotalaria*, *Desmodium*, *Macroptilium*, *Medicago*, and *Mimosa* that invade open agricultural and disturbed habitats [7, 8].

Two of the most important endemic legumes that occur on all the main Hawaiian Islands are trees: *Acacia koa* and *Erythrina sandwicensis*. Although *A. koa* can be found widespread throughout low to high elevation, it dominates canopies in the low mesic to high altitude (900–1800 m) montane rainforests [9, 10]. As volcanic soils are deficient in nitrogen [11], *A. koa* significantly contribute to soil nitrogen cycling by providing nitrogen-rich foliage to the litter layer, support many threatened and endangered bird species, and serve as economically important timber [12]. It is actively being planted in many forest habitat restoration projects [13, 14] with the use of a species of *Bradyrhizobium* as a seedling nursery inoculant [15]. The second species, *Erythrina sandwicensis*, *wiliwili* in Hawaiian, is a keystone species of the dry lowland forest ecosystem across all of the major Hawaiian Islands but is classified as threatened due to tremendous pressure from multiple fronts [16]. Current restoration efforts generally do not use rhizobia inoculants that could provide multiple benefits to *E. sandwicensis* seedlings during the nursery stage and beyond in the field.

Currently, there remains a gap in our knowledge regarding the identity and distribution of rhizobia species in Hawaiʻi, leguminous plants that they associate within, and the soils that they reside. Fine-scale molecular characterization of rhizobia and their associations with hosts is the first step to understanding the ecology of native legumes and their associated symbionts, and aid in the restoration efforts of native legume species. This may include identification of associated rhizobia species or strains, and monitoring of these associations during host establishment in the field. Characterization of the soil is also crucial to develop an understanding of the potential inoculum in that soil, for both restoration and agriculture. Here, we use molecular and culture methods to identify rhizobia genera associated with soils and within nodules of native and non-native legumes across the island of Oʻahu. Then we inoculated selected strains onto *E. sandwicensis* seedlings to show their potentials as nursery inoculants for this threatened species.

## Material and methods

### Sampling and isolation

Soil and legume nodules were sampled across the island of Oʻahu, Hawaiʻi with permission from landowners (Fig 1). The sampling location and site characteristics are shown listed in S1 Table. At each site, at least 3 soil samples were haphazardly sampled from the top 15 cm and combined to make one sample for a total of 377 samples. The soil samples come from diverse soil environments, some of which may not necessarily have leguminous plants actively growing. For sites that have leguminous plants, we targeted a diversity of species, and when possible, multiple plants of the same species were sampled. For each plant, the flowers, leaves, and pods were photographed for identification. Roots of the collected plants were washed thoroughly, and nodules were separated from the larger rooting system and washed until all

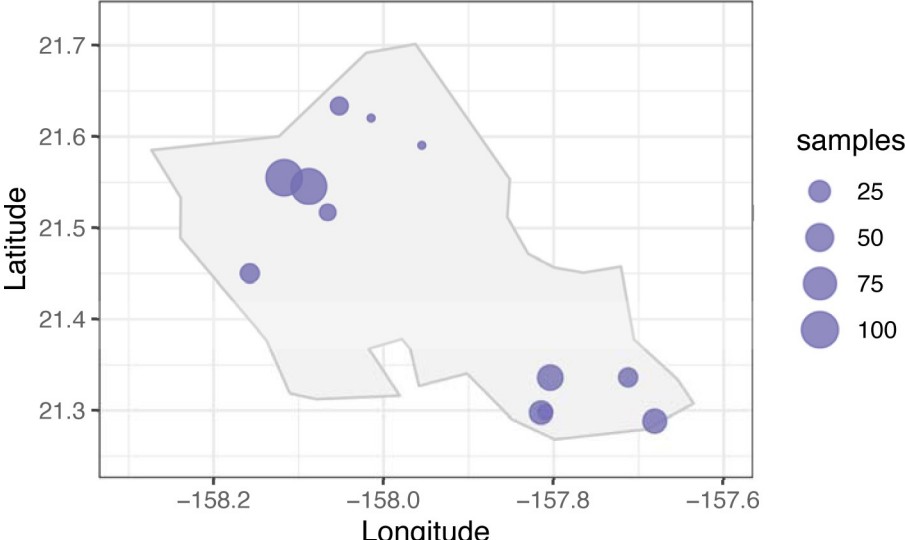

**Fig 1. Map of sampling locations across the island of Oʻahu, Hawaiʻi.** The size of the dots indicates the number of samples collected and sequenced at that site.

apparent soil had been removed. Both soil and nodules were transported back to the laboratory for further processing.

For rhizobia isolation, one root nodule from each plant was washed in 95% ethanol for seven seconds, then soaked in a 2.5% sodium hypochlorite (30% bleach solution) with several drops of Tween 20 for 3 minutes and washed in five successive baths of sterile water. The surface-sterilized root nodules were crushed in 1 mL of sterile water and the nodule solution was then diluted serially to $10^{-6}$ in water. 100 μL of $10^{-6}$ diluted root nodule solution was plated on yeast mannitol agar (YMA, 1 g yeast extract, 10 g mannitol, 0.5 g potassium phosphate dibasic, 0.2 g magnesium sulfate, 0.1 g sodium chloride, and 30 g agar, 1 L water) and incubated for at least 8 days at 27˚C. All morphologically distinct colonies from each nodule were further purified and maintained on YMA.

## Molecular methods

Amplicon library preparation and sequencing followed standard protocols [17]. Briefly, soil samples were homogenized and 0.25 g was used to extract DNA using the PowerSoil Max kit (Qiagen, USA). The amplicon libraries were prepared using a two-step, dual barcoding method [17], using the Earth Microbiome Project primer pairs 515F [18] and 806R [19]. The libraries were sequenced with Illumina MiSeq. The DNA sequences were analyzed using QIIME2 [20] for data quality control following our established pipeline [21]. The ASVs were identified using the SILVA v138 database [22], and the known genera of rhizobia [23] were separated from the main dataset for further analyses.

To identify the cultures, DNA was extracted from actively growing cells using a standard Cetyltrimethylammonium bromide (CTAB) buffer/chloroform extraction protocol and amplified using the primer pair 515FB (5ʹ GTGYCAGCMGCCGCGGTAA 3ʹ) [18] and 926RB (5ʹ CCGYCAATTYMTTTRAGTTT 3ʹ) [19] that target the V4-V5 variable region of the 16S rRNA gene using the following PCR parameters: 95˚C for 30s, 95˚C for 30s, 49˚C for 30 s, 68˚C for 60s, repeat steps 2–4 34x, 68˚C for 300 s. PCR products were cleaned using solid phase

reversible immobilization (SPRI) magnetic beads prior to sequencing using Sanger technology. The resulting chromatograms were manually edited for correct base calls and sequences were clustered at the 99% sequence similarity using the online CD-HIT Suite [24] and representative sequences of each cluster were identified using BLAST that matches to GenBank's curated type strain database.

## Inoculation assay

To determine the efficiency of rhizobia inoculations on the endemic and threatened tree *E. sandwicensis*, we selected two different *Bradyrhizobium* strains that were found with this host: *Bradyrhizobium* sp. strain JA1 (isolate UHC282) is a generalist found across the island of Oʻahu across multiple hosts, and *Bradyrhizobium* strain JA9 (isolate UHC283) was found only in nodules associated with *E. sandwicensis*. A nutrient-poor Oxisol, Wahiawa Series soil collected from the Poamoho Research Station (21.5750803˚, -158.10241˚), was sieved using a 1 mm mesh, mixed 1:1 (v/v) with perlite to improve drainage, and autoclave sterilized at 121˚C for 30 minutes twice, with thorough mixing in between. Fresh cultures of the two isolates above were suspended in double deionized water and immediately inoculated onto the soil to achieve $10^6$ rhizobia cells/mL of soil. Sterile plastic pots were filled with 480 mL of inoculated soil, or sterile uninoculated soil watered with 5 mL of 0.5% $KNO_3$ weekly as a positive control and watered with tap water as a negative control. Seeds were surface sterilized and scarified, prior to planting into each pot, and grown for 10 weeks under greenhouse conditions with daily watering. There were five replicates per treatment (N = 20).

To measure inoculum efficiency, we used a combination of several non-destructive metrics that allowed us to preserve these threatened plants. Stem height was measured from the soil line of the plant to the tip apical meristem; approximate stem volume was calculated using the formula for cone volume $V = (1/3)\pi r^2 h$ using basal stem radius (r) and height (h). The total number of nodules was counted, and some were cut open to confirm active nodulation through color. Wet weight of the whole seedling was taken by first washing the root system clean of soil and dabbed dry prior to weighing, taking care to perform these steps as consistently as possible. Together, these data provided a general measure of inoculum efficiency of the inoculated rhizobia strains [25].

## Statistical analyses

All statistical analyses were conducted by using R [26]. The sampling map was drawn using the package *ggmap* [27]. For each sample in the soil dataset, the sequence data were condensed at the genus level, and the data was collapsed to presence and absence. The data was then normalized across each sampling site by percentage. In other words, the prevalence of taxa in this study was based on its presence relative to total number of samples. This prevalence was plotted as a "bubble chart" using the *ggplot2* package [28]. To measure whether there is differential occurrence between site disturbance, we used the "adonis" function in the *vegan* package [29]. The mean differences between *E. sandwicensis* growth were conducted via the Least Significant Distance test with a 0.95 confidence interval with the *agricolae* package [30], and graphs were plotted by using the *ggplot2* package [28]. When calculating the averages for each experimental group, the seedlings that did not have nodules were not incorporated. Only two of the seedlings in the positive (fertilized) control survived, and thus this treatment was removed from further analyses.

## Results

### Rhizobia from soil sequences

In this study, we identified 18 genera of rhizobia on Oʻahu from soils and nodules (Fig 2). Some genera could not be separated from each other (e.g. the *Rhizobium* complex = *Allorhizobium-Neorhizobium-Pararhizobium-Rhizobium;* the *Burkholderia* complex = *Burkholderia-Paraburkholderia;* the *Methylobacterium* complex = *Methylobacterium-Methylorubrum*) based on the short V4 segment of the 16S rRNA gene. We note that it was not possible to clearly detect which members of *Burkholderia* and the *Methylobacterium* complex nodulate legumes based on the sequence data, but they have been included for completeness. Therefore, we report 13 operational genera. Of these, the *Rhizobium* complex, *Bradyrhizobium*, *Devosia*, *Mesorhizobium*, and *Microvirga* appear to be the most widespread, but only *Cupriavidus* was found across all sites sampled, although often at lower abundance than other genera.

Although the rhizobia genera do not seem to cluster according to sites, soil or land-use, agricultural sites tend to have most, if not all genera, whereas non-agricultural sites tend to have fewer (Fig 2). Of note is the Waimea Ridge site that is only accessible through helicopter. This site has the lowest genus richness with only *Bradyrhizobium*, the *Burkholderia* complex, *Cupriavidus*, and the *Methylobacterium* complex. There was no significance difference (p = 0.545) between the genera that occurred in agricultural soils compared to less disturbed or natural soils.

### Rhizobia isolates from nodules

We cultured 55 rhizobia isolates from 24 different legume species (Table 1). The majority of isolates (62%) were sampled from the Poamoho Research Station site. The isolates were identified as *Bradyrhizobium* (30 isolates, 3 strains), *Ensifer (Sinorhizobium*, 10 isolates, 3 strains), *Rhizobium* (6 isolates, 2 strains), *Mesorhizobium* (2 isolates, 1 strain), *Cupriavidus* (6 isolate, 1 strain), and *Bosea* (1 isolate, 1 strain). The most common strain, *Bradyrhizobium* strain JA1 (23 isolates, 98.92% match to *Bradyrhizobium elkanii*) was found with nine legume species, including *E. sandwicensis* and *A. koa*, the two endemic legume species sampled in this study. The other notable strains were *Bradyrhizobium* strain JA2 (98.92% match to *Bradyrhizobium vignae*), *Ensifer* strain JA3 (99.18% match to *Ensifer alkalisoli/medicae* complex), *Rhizobium* strain JA4 (99.18% match to *Rhizobium indigoferae*), and *Cupriavidus* strain JA10 (99.52% match to *Cupriavidus oxalaticus*) came from diverse hosts.

The overwhelming majority of individual plants associated with multiple rhizobia strains, whether they were collected from the same location (e.g. *Arachis pintoi*) or from different locations (e.g. *Crotalaria juncea, Leucaena leucocephala*) (Table 1). It is worth noting that the two endemic plant species *E. sandwicensis* and *A. koa* were both associated with *Bradyrhizobium* sp. strain JA1, but in addition, *E. sandwicensis* was associated with *Bradyrhizobium* sp. strain JA9. Generally, an individual nodule contains only one isolate of rhizobia (although they can contain other bacteria). We found that in two cases, a single nodule contained two different rhizobia strains. An *Arachis pintoi* nodule contained *Bradyrhizobium* JA1 and *Rhizobium* sp. strain JA4, and a *Desmodium tortuosum* nodule, contained both *Rhizobium* sp. strain JA4 and *Agrobacterium* sp. strain JA13.

### Inoculation assays

Inoculation of seedlings using the cell slurry method successfully produced nodules and had positive effects on the host in certain measurements (Fig 3). 80% of inoculated plants formed nodules, although the number of nodules was relatively few and tended to cluster close the

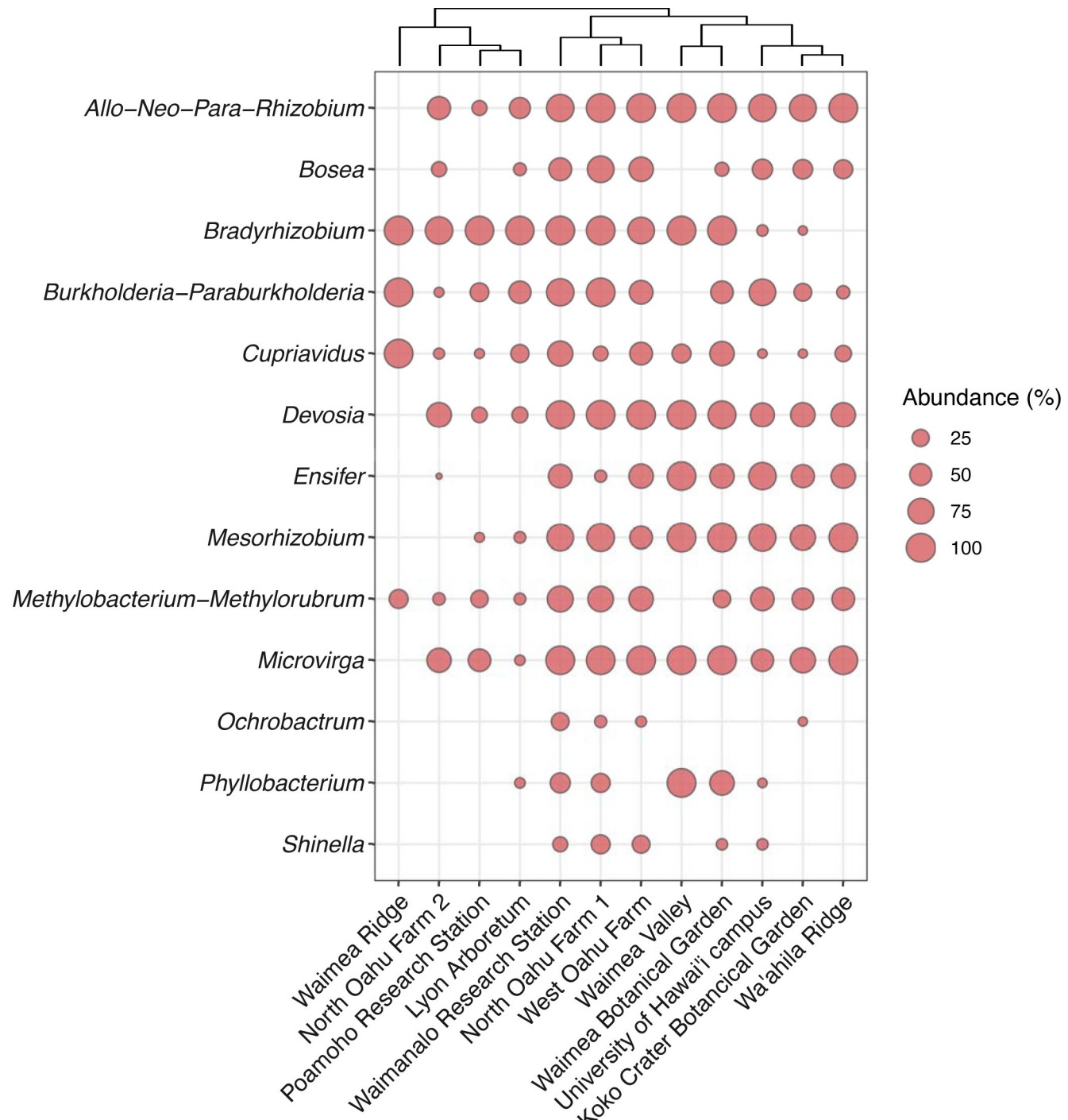

**Fig 2. Genera of plant-nodulating rhizobia that occur on the island of Oʻahu, Hawaiʻi.** The presence of each genus (or genus complex) is indicated by a dot where the size corresponds to the normalized abundance. The sites are clustered based on the presence of genera and their normalized abundance.

main taproot. There was no significant number of nodules formed between the two isolates. Nodulated plants produced noticeably larger root systems compared to the control. Although we did not quantify the density of these roots, the contribution of the larger inoculated root system likely contributed to the overall significant increase in plant wet weight biomass

**Table 1. Bacteria strains isolated from legume nodules of diverse hosts on the island of Oʻahu, Hawaiʻi.**

| Strain | Host Species | Locality | Isolate | GenBank Acession | Best BLAST to Type# | % Match to Type |
|--------|-------------|----------|---------|-----------------|---------------------|-----------------|
| JA1 | *Acacia koa* | Camp Palehua | UHC275 | MT703739 | *Bradyrhizobium elkanii* | 98.92% |
| JA1 | *Acacia koa* | Camp Palehua | UHC276 | MT703740 | | |
| JA1 | *Acacia koa* | Camp Palehua | UHC277 | MT703741 | | |
| JA1 | *Acacia koa* | Camp Palehua | UHC278 | MT703742 | | |
| JA1 | *Arachis pintoi* | Poamoho Station | UHC271 | MT703735 | | |
| JA1 | *Arachis pintoi* | Poamoho Station | UHC274* | MT703738 | | |
| JA1 | *Canavalia sericea* | Waimanalo Beach | UHC251 | MT703716 | | |
| JA1 | *Centrosema pubescens* | Poamoho Station | UHC240 | MT703705 | | |
| JA1 | *Chamaecrista nictitans* | Poamoho Station | UHC269 | MT703733 | | |
| JA1 | *Chamaecrista nictitans* | Poamoho Station | UHC270 | MT703734 | | |
| JA1 | *Chamaecrista nictitans* | Poamoho Station | UHC273 | MT703737 | | |
| JA1 | *Crotalaria juncea* | Waialua Farms | UHC250 | MT703715 | | |
| JA1 | *Crotalaria pallida* | Poamoho Station | UHC235 | MT703700 | | |
| JA1 | *Crotalaria spectabilis* | Poamoho Station | UHC226 | MT703691 | | |
| JA1 | *Erythrina sandwicenesis* | Koko Crater BG | UHC282 | MT703746 | | |
| JA1 | *Indigofera spicata* | Poamoho Station | UHC227 | MT703692 | | |
| JA1 | *Indigofera suffruticosa* | Poamoho Station | UHC254 | MT703719 | | |
| JA1 | *Macroptilium lathyroides* | Poamoho Station | UHC236 | MT703701 | | |
| JA1 | *Pachyrhizus erosus* | Poamoho Station | UHC237 | MT703702 | | |
| JA1 | *Pachyrhizus erosus* | Poamoho Station | UHC263 | MT703728 | | |
| JA1 | *Pachyrhizus erosus* | Poamoho Station | UHC267 | MT703731 | | |
| JA1 | *Psophocarpus tetragonolobus* | Waimanalo Station | UHC247 | MT703712 | | |
| JA1 | *Vigna unguiculata* | Poamoho Station | UHC230 | MT703695 | | |
| JA2 | *Desmodium tortuosum* | Waimanalo Station | UHC246 | MT703711 | *Bradyrhizobium vignae* | 98.92% |
| JA2 | *Indigofera suffruticosa* | Poamoho Station | UHC225* | MT703690 | | |
| JA2 | *Pachyrhizus erosus* | Poamoho Station | UHC272 | MT703736 | | |
| JA2 | Unidentified sp. 1 | Poamoho Station | UHC252 | MT703717 | | |
| JA2 | Unidentified sp. 2 | Poamoho Station | UHC268 | MT703732 | | |
| JA2 | Vigna sp. 1 | Waimanalo Station | UHC245 | MT703710 | | |
| JA3 | *Chamaecrista nictitans* | Poamoho Station | UHC231 | MT703696 | *Ensifer alkalisoli/medicae* | 99.18% |
| JA3 | *Chamaecrista nictitans* | Poamoho Station | UHC232 | MT703697 | | |
| JA3 | *Medicago sativa* | Poamoho Station | UHC234 | MT703699 | | |
| JA3 | *Medicago sativa* | Poamoho Station | UHC255 | MT703720 | | |
| JA3 | *Medicago sativa* | Poamoho Station | UHC256 | MT703721 | | |
| JA3 | *Medicago sativa* | Poamoho Station | UHC258* | MT703723 | | |
| JA3 | Unidentified sp. 2 | Poamoho Station | UHC233 | MT703698 | | |
| JA4 | *Arachis pintoi* | Poamoho Station | UHC261 | MT703726 | *Rhizobium indigoferae* | 99.18% |
| JA4 | *Arachis pintoi* | Poamoho Station | UHC262 | MT703727 | | |
| JA4 | *Desmodium tortuosum* | Waimanalo Station | UHC248 | MT703713 | | |
| JA4 | *Leucaena leucocephala* | Poamoho Station | UHC259 | MT703724 | | |
| JA4 | *Phaseolus vulgaris* | Waimanalo Station | UHC243* | MT703708 | | |
| JA5 | *Leucaena leucocephala* | Waimanalo Station | UHC242 | MT703707 | *Ensifer glycinis* | 98.92% |
| JA5 | *Leucaena leucocephala* | UH Manoa Campus | UHC253* | MT703718 | | |
| JA6 | *Leucaena leucocephala* | Poamoho Station | UHC257* | MT703722 | *Mesorhizobium acaciae* | 99.18% |
| JA6 | *Leucaena leucocephala* | Poamoho Station | UHC260 | MT703725 | | |
| JA8 | *Crotalaria juncea* | Poamoho Station | UHC238* | MT703703 | *Ensifer adhaerens* | 99.18% |
| JA9 | *Erythrina sandwicenesis* | Koko Crater BG | UHC283* | MT703747 | *Bradyrhizobium betae* | 98.88% |

(*Continued*)

**Table 1.** (Continued)

| Strain | Host Species | Locality | Isolate | GenBank Acession | Best BLAST to Type[#] | % Match to Type |
|---|---|---|---|---|---|---|
| JA10 | *Mimosa pudica* | Waimanalo Station | UHC241 | MT703706 | *Cupriavidus oxalaticus* | 99.52% |
| JA10 | *Mimosa pudica* | O'ahu | UHC533 | OM980333 | | |
| JA10 | *Mimosa pudica* | Wa'ahila Ridge | UHC539 | OM980334 | | |
| JA10 | *Mimosa pudica* | Wa'ahila Ridge | UHC540* | OM980335 | | |
| JA10 | *Mimosa pudica* | UH Manoa Campus | UHC541 | OM980336 | | |
| JA10 | *Mimosa pudica* | UH Manoa Campus | UHC542 | OM980337 | | |
| JA11 | *Neonotonia wightii* | Poamoho Station | UHC229* | MT703694 | *Rhizobium alamii* | 98.89% |
| JA12 | *Crotalaria juncea* | Poamoho Station | UHC536 | OM980339 | *Bosea robiniae* | 99.77% |

Endemic host species are in bolded font. The asterisk

(*) indicates the representative isolate for a strain, determined as the best sequence for that strain by the CD-HIT software.

[#] The BLAST results show the best match to that representative sequence to the most closely related Type strain. BG = Botanical Garden.

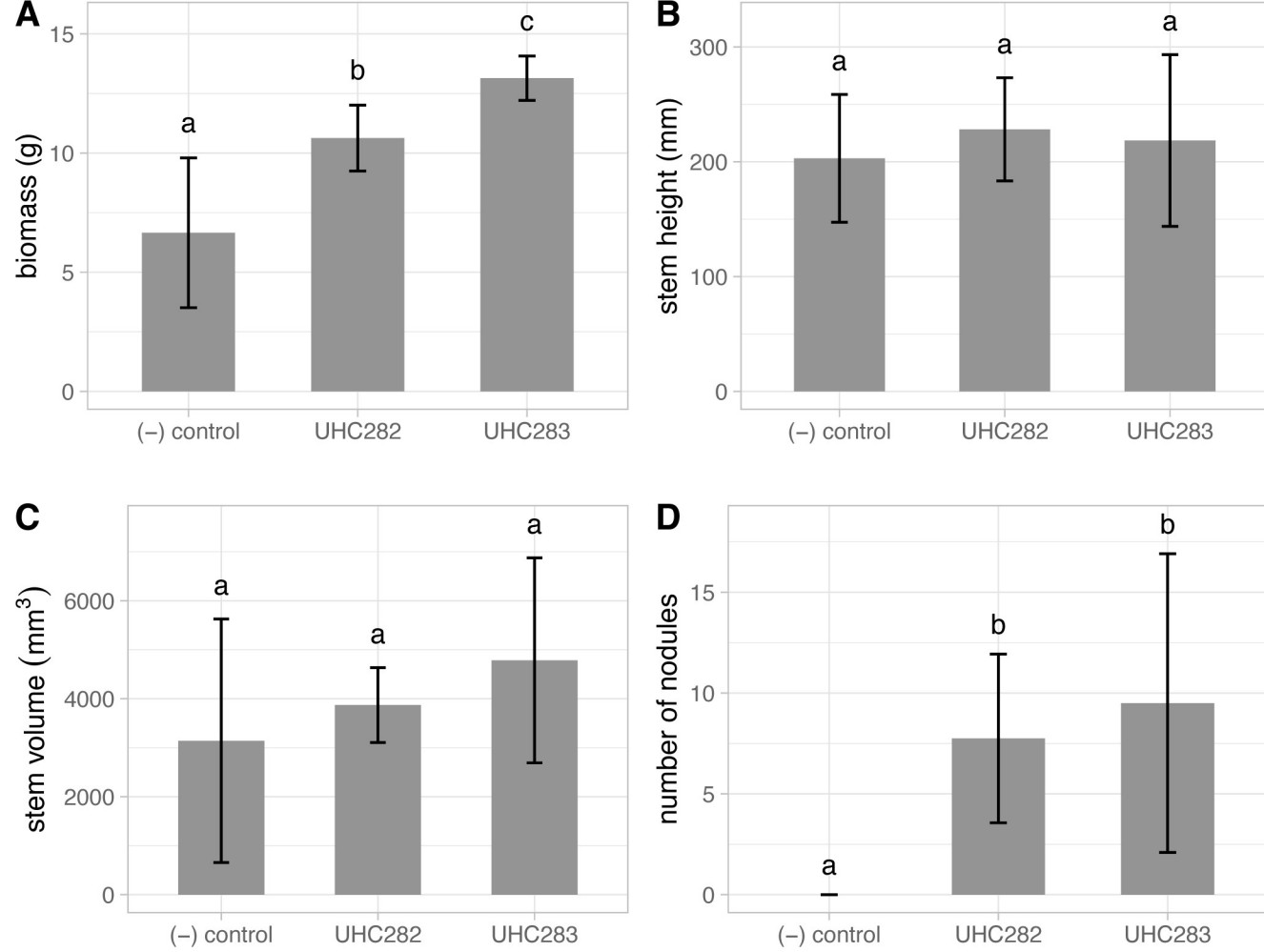

**Fig 3.** ***Erythrina sandwicensis* seedling response to inoculation with *Mesorhizobium* spp.** Isolate UHC282 (strain JA1) and *Bradyrhizobium* sp. isolate UHC283 (strain JA9). A) wet biomass, B) stem height, C) stem volume, and D) number of nodules. Letters above the standard error bars represent significance differences.

compared to the control (p < 0.001). Colonization by isolate UHC283 (strain JA9) significantly increased total wet plant biomass compared to isolate UHC282 (strain JA1) and the control. There were no significant differences between stem height (p = 0.662) and stem volume (p = 0.299) of either isolate compared to each other and the negative control.

## Discussion

In this study, we characterized plant-nodulating rhizobia from 12 sites across the island of Oʻahu, Hawaiʻi from both soil and active nodules and found that all plant-nodulating rhizobia genera are present on Oʻahu, although only a subset of these genera were found forming nodules. We did not detect any discernable patterns in the distribution of these genera, suggesting that they are ubiquitous in disturbed habitats. Of these habitats, agricultural sites tend to have the largest number of genera compared to the undisturbed Waimea Ridge site that has the fewest genera. This observation is consistent with our understanding of the history of agriculture and introduction of microbes to agricultural sites, but the lower rhizobia genera richness reflects the lower richness of native Hawaiian plant species in intact habitats.

Of the five most common genera, *Cupriavidus* can be found at all sites but only *Bradyrhizobium* can nodulate a variety of host plants. *Cupriavidus* (Order Burkholdariales) were isolated exclusively from *Mimosa pudica* nodules and is consistent with their preference for only *Mimosa* species [31]. Here we found the genus to be present in the broad array of soils from nutrient poor agricultural soils to undisturbed native soils, and soils with thick fungal mycelium mats. Many of these soils do not have *M. pudica* plants, but there is evidence showing that *Cupriavidus necator* can nodulate various leguminous species [32, 33], including *Leucaena leucocephala* that is widely distributed across Oʻahu, with the exception of the more undisturbed sites. In contrast, the most common strain that associates broadly with many hosts was *Bradyrhizobium* JA1. These findings reflect other works showing that *Bradyrhizobium* species are ubiquitous in tropical soils due to their ability to nodulate a broad range of tropical legumes [9, 34, 35]. We also isolated *Ensifer*, *Mesorhizobium*, and *Rhizobium*, but these occurred in lower frequency than *Bradyrhizobium*, although they are also widespread. The exceptions are *Devosia* and *Microvirga* (Order *Hyphomicrobiales*), which we were not able to isolate, perhaps due to their low level of occurrence. The occurrence of *Cupriavidus* and other rhizobia such as *Mesorhizobium* in the rhizosphere of many non-legume plant hosts and fungal-rich habitats suggests that they have an external "free living" phase in the rhizosphere where they consume root and fungal exudates [36–38], and a more well-known symbiotic phase inside of nodules. A much more detailed sampling and species-level identification these rhizobia strains will be required to confirm these associations.

We observed an occurrence of multiple rhizobia strains within a single nodule ("mixed nodulation") of the ground peanut *Arachis pintoi*. The nodule contained *Bradyrhizobium* sp. strain JA1 and *Rhizobium* sp. strain JA4. This phenomenon of mixed nodulation is unusual because through host sanctioning, legumes preferentially select strains that have higher rates of nitrogen fixation. Certain rhizobia species are inefficient at fixing nitrogen or cannot fix nitrogen; thus, these rhizobia would be sanctioned from nodulating the respective legume [39]. However, mixed nodulation of different strains with varying nitrogen fixation efficiencies does not necessarily impact the plant negatively [40]. Therefore, less efficient rhizobia strains have evolved mechanisms that allow them to escape sanction and cohabit the same nodule with efficient strains. The fact that we have discovered multiple strains associating with a single nodule with limited sampling suggests that this occurrence is more common in nature than previously thought and deserves further attention in future works.

We showed that soil inoculations with fresh cultures, followed by planting *E. sandwicensis* seeds, is a relatively effective way to introduce rhizobia to seedlings in a nursery setting. At the soil inoculation rate of $1\times10^6$ cells/ml of soil, only 80% of seedlings were successfully nodulated. Various factors could have contributed to this, including the deactivation active cells prior to seedlings forming colonizable roots. The time of inoculation to germination of *E. sandwicensis* usually ranges about 5–10 days. Perhaps this time delay was enough to render enough rhizobia cells dead or inactive so that they are no longer able to nodulate seedlings. Seedlings that were successfully inoculated had significantly larger biomass. Since there were no significant differences in stem height and volume, the difference was likely driven by a larger and denser root system. Measurements using dry weight would have been the best practice, but we chose to preserve these threatened plants and found that a combination of various measurements provided acceptable data to distinguish the treatments apart [25]. We found that inoculation using *Mesorhizobium* strain JA1 wasn't as effective as the less common *Bradyrhizobium* strain JA9. This indicates that perhaps strain JA9 is more specific to *E. sandwicensis*, although much more rigorous sampling will be required to confirm this relationship.

Our understanding of the patterns of rhizobia and host associations are limited in this study because of several factors. Although our soil dataset allowed some inferences into the occurrence of rhizobia across the island, our modest sampling of host-connected nodules and the low resolution of 16S rRNA gene did not allow us to make strong conclusions about the specific patterns of host and symbiont associations, nor did they provide enough insights into whether some strains were native or introduced. However, as Hawaiʻi is the "invasive capital" of the world, it is likely that many of the strains we found with non-native plants also came by ways of human introductions. What is clear, however, is that the NifTAL project brought in many non-native strains to Hawaiʻi [5] and these strains likely still exist, especially in agricultural soils such as the ones we sampled in this study. The strains isolated from the two species of endemic plants came from somewhat disturbed habitats, and thus it is difficult to determine whether they are truly native strains. On the island of Oʻahu where we sampled, the few intact habitats left are found in upper elevation ridges with relatively minor disturbance from anthropogenic activities. Future sampling of native legumes from these undisturbed habitats will be more meaningful to isolate native strains, as there is a general pattern of native microbes occurring with native plants, exemplified by some species of mushroom-forming fungi in native forest habitats [41].

## Conclusions

In this study, we showed that the soils across the island of Oʻahu contain diverse rhizobia, some of which can be isolated from legume nodules. We gained insights into some general patterns such as the richness of rhizobia genera, and the generalist strains that occur widely across our sampling sites. This suggests that there is a constant source of rhizobia inoculants in these soils, and cultivation of leguminous plants do not need inoculants, except for those that associate with specific strains of rhizobia. Using strains that were isolated from *E. sandwicensis*, we showed that *E. sandwicensis* seedlings can be inoculated with fresh cultures, resulting in seedlings with significantly larger biomass. Although both strains resulted in positive seedling growth, there appears to be differences in plant biomass, so therefore we recommend that seedlings be inoculated with *Bradyrhizobium* strain JA9 in the nursery before transplanting into the ground. Future research to test delivery systems, such as applying lyophilized inoculum directly into the soil, mixing directly into water and applying to the medium surface with growing plants, or coating of seeds for field inoculations will be helpful to the restoration efforts of the threatened and endangered legumes in Hawaiʻi.

## Supporting information

**S1 Table. Site location and characteristics of the soil samples in this study.** Samples at each site was obtained with permission from landowners. The GPS coordinates at these sites are not reported to protect the identity of each farm as per our sampling agreement.
(DOCX)

## Acknowledgments

We thank Jesse Mikasobe-Kealiinohomoku for enthusiastic help with fieldwork collecting nodules from native legumes, and Michael Muszynski for guidance to JNAA during his development as an undergraduate researcher. JNAA thanks Forest and Kim Starr for assistance in identification of some legume host plants.

## Author Contributions

**Conceptualization:** Nhu H. Nguyen.

**Data curation:** Jonathan N. A. Abe.

**Formal analysis:** Jonathan N. A. Abe.

**Funding acquisition:** Jonathan N. A. Abe, Nhu H. Nguyen.

**Investigation:** Jonathan N. A. Abe, Ishwora Dhungana.

**Methodology:** Jonathan N. A. Abe, Ishwora Dhungana, Nhu H. Nguyen.

**Project administration:** Nhu H. Nguyen.

**Resources:** Nhu H. Nguyen.

**Validation:** Jonathan N. A. Abe, Nhu H. Nguyen.

**Visualization:** Nhu H. Nguyen.

**Writing – original draft:** Jonathan N. A. Abe, Nhu H. Nguyen.

**Writing – review & editing:** Jonathan N. A. Abe, Ishwora Dhungana, Nhu H. Nguyen.

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
