## [Decision Letter · Decision Letter 0]

4 Jul 2023

PONE-D-23-15692Legume-nodulating rhizobia are widespread in soils and plants across the island of O‘ahu, Hawai‘iPLOS ONE

Dear Dr. Nguyen,

Thank you for submitting your manuscript to PLOS ONE. After careful consideration, we feel that it has merit but does not fully meet PLOS ONE’s publication criteria as it currently stands. Therefore, we invite you to submit a revised version of the manuscript that addresses the points raised during the review process.

We look forward to receiving your revised manuscript.

Kind regards,

Ying Ma, Ph.D.

Academic Editor

PLOS ONE

Journal Requirements:

Reviewers' comments:

Reviewer's Responses to Questions

**Comments to the Author**

1. Is the manuscript technically sound, and do the data support the conclusions?

Reviewer #1: Yes

Reviewer #2: Yes

2. Has the statistical analysis been performed appropriately and rigorously? 

Reviewer #1: Yes

Reviewer #2: N/A

3. Have the authors made all data underlying the findings in their manuscript fully available?

Reviewer #1: Yes

Reviewer #2: No

4. Is the manuscript presented in an intelligible fashion and written in standard English?

Reviewer #1: Yes

Reviewer #2: Yes

5. Review Comments to the Author

Reviewer #1: In my opinion, the manuscript is well presented and with interesting results. My suggestion would be to prove that in fact 2 different bacteria can be in the same nodule and further amplify, sequence and analyze some nodulation gene to explain the nodulation variability of bacteria isolated from nodules. This result could even demonstrate which bacteria originated from other regions or from Hawaii.

Reviewer #2: This manuscript presents an interesting piece of research in the field of rhizobiology that tries to characterize root nodulating isolates of the rhizobia complex from soils and plants in the island of O'ahu, Hawai. The manuscript is worth reading particularly when it comes to sustainability in the sense of the beneficial impacts of biological nitrogen fixation on the restoration of indigenous legumes as well as in soil health and legume crop productivity. However, the manuscript fails to present the anticipated outcome of the initially planned target. Below are a few of the most crucial comments that needs to be addressed for this manuscript to be published on PLOs.

1. Of all the isolates obtained from such a large group of legumes, what is the basis for choosing or limiting yourself to only two isolates for the nodulation inoculation study?

2. Can the authors please elaborate in your introduction how molecular characterization of rhizobia important in the restoration of native legumes?

3. The various legumes species indicated in Table 1 undoubtedly have different requirement, growth stage, seasonal requirement for growth as well as nodulation process. Hence, it is very difficult to say that all of the studied legumes would reach their active nodulation or growth stage all at the same time, say during the time of sample collection. Some of the legumes might be at their active nodulation stage, while a number of others might wither away their nodules or become senescence. So, how were the authors able to sample active nodules all the same time for all the different species of native legumes?

4. How the concentration of the initial inoculum adjusted before introduced into the soil? How was the inoculum introduced and at what rate?

5. No clear information was provided about the experimental layout, the treatments and their replication.

6. How was evaluation for nodulation performed 10 weeks after planting and inoculation, while most legumes become actively nodulating around the 6th to 7th week? Is this the same for all the legumes tested?

7. I am wondering while in the introductory section the authors mentioned about the role of rhizobia or the legume rhizobium symbiosis in the restoration of native legumes, no effort was made to evaluate the inoculated rhizobia for their nitrogen fixation ability. Even the nodulation evaluation was not sufficient as it could have been wise to include such parameters as nodule dry weight, nodule positions on the root as well as color of the nodules.

8. Table 1. What do you mean by 'the representative sequence of the strain'? Do you mean to refer to 'type strains', if so how was it determined?

9. In the materials and methods, please indicate the method (and citation) used to generate the results in Figure 2.

10. Why was not a phylogenetic analysis made to elucidate the evolutionary relatedness and taxonomic delineation of the rhizobia isolated which could also be used to study how these legumes made adaptations to be the microbial symbionts of their respective legumes.

11. No result is provided for the amplicon sequencing from the soil despite describing the library preparation and Illumina sequencing techniques. These results need to be included in the manuscript including their interpretation and discussion.

6. PLOS authors have the option to publish the peer review history of their article (what does this mean?). If published, this will include your full peer review and any attached files.

Reviewer #1: No

Reviewer #2: **Yes: **Ahmed Idris Hassen

---

## [Author Response · Author response to Decision Letter 0]

20 Jul 2023

Response to Reviewers' comments:

Reviewer #1: In my opinion, the manuscript is well presented and with interesting results. My suggestion would be to prove that in fact 2 different bacteria can be in the same nodule and further amplify, sequence and analyze some nodulation gene to explain the nodulation variability of bacteria isolated from nodules. This result could even demonstrate which bacteria originated from other regions or from Hawaii.

This is indeed an interesting topic. Unfortunately, the data gathering phase of the project is complete and we do not have the capacity to fulfill this request. We feel that it is beyond the scope of the work that we’d like to present. However, to address this comment, we have added a paragraph to the discussion (Lines 284-295) on this mixed inoculation phenomenon and referenced studies that have already pursued this area of research.

“We observed an occurrence of multiple rhizobia strains within a single nodule (“mixed nodulation”) of the ground peanut Arachis pintoi. The nodule contained Bradyrhizobium sp. strain JA1 and Rhizobium sp. strain JA4. This phenomenon of mixed nodulation is unusual because through host sanctioning, legumes preferentially select strains that have higher rates of nitrogen fixation. Certain rhizobia species are inefficient at fixing nitrogen or cannot fix nitrogen; thus, these rhizobia would be sanctioned from nodulating the respective legume [39]. However, mixed nodulation of different strains with varying nitrogen fixation efficiencies does not necessarily impact the plant negatively [40]. Therefore, less efficient rhizobia strains have evolved mechanisms that allow them to escape sanction and cohabit the same nodule with efficient strains. The fact that we have discovered multiple strains associating with a single nodule with limited sampling suggests that this occurrence is more common in nature than previously thought and deserves further attention in future works.”

Reviewer #2: This manuscript presents an interesting piece of research in the field of rhizobiology that tries to characterize root nodulating isolates of the rhizobia complex from soils and plants in the island of O'ahu, Hawai. The manuscript is worth reading particularly when it comes to sustainability in the sense of the beneficial impacts of biological nitrogen fixation on the restoration of indigenous legumes as well as in soil health and legume crop productivity. However, the manuscript fails to present the anticipated outcome of the initially planned target. Below are a few of the most crucial comments that needs to be addressed for this manuscript to be published on PLOs.

1. Of all the isolates obtained from such a large group of legumes, what is the basis for choosing or limiting yourself to only two isolates for the nodulation inoculation study?

We had limited access to the seeds of this threatened plant, so we had to limit the inoculation portion of this study to just the two strains that were found with this host. The reasoning for selecting these two strains was explained in Lines 133-137:

“To determine the efficiency of rhizobia inoculations on the endemic and threatened tree E. sandwicensis, we selected two different Bradyrhizobium strains that were found with this host: Bradyrhizobium sp. strain JA1 (isolate UHC282) is a generalist found across the island of O‘ahu across multiple hosts, and Bradyrhizobium strain JA9 (isolate UHC283) was found only in nodules associated with E. sandwicensis.”

2. Can the authors please elaborate in your introduction how molecular characterization of rhizobia important in the restoration of native legumes?

Thank you for this suggestion. We have added the following to the introduction, Lines 72-75:

“This may include identification of host-associated rhizobia species or strains, followed by monitoring of these associations during host establishment. Characterization of the soil is also crucial to develop an understanding of the potential inoculum in that soil, for both restoration and agriculture.”

3. The various legumes species indicated in Table 1 undoubtedly have different requirement, growth stage, seasonal requirement for growth as well as nodulation process. Hence, it is very difficult to say that all of the studied legumes would reach their active nodulation or growth stage all at the same time, say during the time of sample collection. Some of the legumes might be at their active nodulation stage, while a number of others might wither away their nodules or become senescence. So, how were the authors able to sample active nodules all the same time for all the different species of native legumes?

We agree with this assessment that perhaps not all nodules were active when sampled. However, we did not claim to have sampled active nodules. We sampled the soils that have plants actively growing. To avoid any confusion, we have clarified the sentence to indicate that the activity was from the plant, and not the nodules in Lines 85-86:

“The soil samples come from diverse soil environments, some of which may not 

have leguminous plants actively growing.”

4. How the concentration of the initial inoculum adjusted before introduced into the soil? How was the inoculum introduced and at what rate?

We have reported this in Lines 140-142: 

“Fresh cultures of the two isolates above were suspended in double deionized water and immediately inoculated onto the soil to achieve 106 rhizobia cells/mL of soil.”

5. No clear information was provided about the experimental layout, the treatments and their replication.

We are unsure what the review meant by experimental layout. How we treated the plants, inoculated the soils, and the replication was reported in Lines 137-146. We have added the total number of plants to try and clarify this confusion in Lines 145-146:

“There were five replicates per treatment (N=20).”

6. How was evaluation for nodulation performed 10 weeks after planting and inoculation, while most legumes become actively nodulating around the 6th to 7th week? Is this the same for all the legumes tested?

We did observe nodulation after 6 weeks, but chose to wait until 10 weeks to make sure that we gave all the plants plenty of time to be nodulated in case there are latencies in development that we did not anticipate. Because this is a host that has not been studied for nodulation before, we decided to be conservative in our time. We sampled all the greenhouse inoculated legumes at the same time at 10 weeks.

7. I am wondering while in the introductory section the authors mentioned about the role of rhizobia or the legume rhizobium symbiosis in the restoration of native legumes, no effort was made to evaluate the inoculated rhizobia for their nitrogen fixation ability. Even the nodulation evaluation was not sufficient as it could have been wise to include such parameters as nodule dry weight, nodule positions on the root as well as color of the nodules.

We have chosen to use the biomass of the plant as a proxy for active nitrogen fixation. Since our soil was nitrogen poor, any significant growth that came from the plant must have come from fixation. This was shown in Figure 1A where uninoculated plants were significantly smaller than inoculated plants. We have added our observations of the color of the nodules in Lines 151-152: 

“The total number of nodules was counted, and some were cut open to confirm active nodulation through color.”

8. Table 1. What do you mean by 'the representative sequence of the strain'? Do you mean to refer to 'type strains', if so how was it determined?

The sentence was indeed confusing. Each strain has multiple isolates, so we picked one to represent the strain. This representative is determined by the CD-HIT software that we used to make multiple comparisons of the strains. It is based on sequence length and the number of sequences that matches exactly to it. We have clarified the table caption in Lines 214-217:

“The asterisk (*) indicates the representative isolate for a strain, determined as the best sequence for that strain by the CD-HIT software. # The BLAST results show the best match to that representative sequence to the most closely related Type strain.”

9. In the materials and methods, please indicate the method (and citation) used to generate the results in Figure 2.

The method and citation were added to lines 163-164:

“This prevalence was plotted as a “bubble chart” using the ggplot2 package [28]”

10. Why was not a phylogenetic analysis made to elucidate the evolutionary relatedness and taxonomic delineation of the rhizobia isolated which could also be used to study how these legumes made adaptations to be the microbial symbionts of their respective legumes.

This would be a nice next step to this study. Currently the limited sampling of any single species, and the short segment of the 16S rRNA gene sequenced does not allow for robust determination of any co-evolutionary relationships.

11. No result is provided for the amplicon sequencing from the soil despite describing the library preparation and Illumina sequencing techniques. These results need to be included in the manuscript including their interpretation and discussion.

Figure 2 shows these data. We have clarified this in Line 175: 

“In this study, we identified 18 genera of rhizobia on O‘ahu from soils and nodules (Fig 2).”

We also noted that the reviewer indicated “no” in the review questionnaire #3. “Have the authors made all data underlying the findings in their manuscript fully available?”. All the raw sequence data have been deposited to GenBank and the Sequence Read Archive (SRA). These data are scheduled to be released on Sept. 1, 2023 or when this manuscript has been published. The scripts used to process those data have been deposited on GitHub. This can be found under the “Availability of data and material” statement on Lines 363-367.

---

## [Decision Letter · Decision Letter 1]

24 Aug 2023

Legume-nodulating rhizobia are widespread in soils and plants across the island of O‘ahu, Hawai‘i

PONE-D-23-15692R1

Dear Dr. Nguyen,

We’re pleased to inform you that your manuscript has been judged scientifically suitable for publication and will be formally accepted for publication once it meets all outstanding technical requirements.

Kind regards,

Ying Ma, Ph.D.

Academic Editor

PLOS ONE

Additional Editor Comments (optional):

Reviewers' comments:

Reviewer's Responses to Questions

**Comments to the Author**

1. If the authors have adequately addressed your comments raised in a previous round of review and you feel that this manuscript is now acceptable for publication, you may indicate that here to bypass the “Comments to the Author” section, enter your conflict of interest statement in the “Confidential to Editor” section, and submit your "Accept" recommendation.

Reviewer #1: All comments have been addressed

Reviewer #2: All comments have been addressed

2. Is the manuscript technically sound, and do the data support the conclusions?

Reviewer #1: Yes

Reviewer #2: Yes

3. Has the statistical analysis been performed appropriately and rigorously? 

Reviewer #1: Yes

Reviewer #2: Yes

4. Have the authors made all data underlying the findings in their manuscript fully available?

Reviewer #1: Yes

Reviewer #2: Yes

5. Is the manuscript presented in an intelligible fashion and written in standard English?

Reviewer #1: Yes

Reviewer #2: Yes

6. Review Comments to the Author

Reviewer #1: As I mentioned in the previous article, the paper brings new information that I believe is important to be published. Every suggestion and question I asked was addressed. Therefore, I recommend publishing the article.

Reviewer #2: The reviewers have made all my comments in the initial review process. The English standard is good, and all typo errors have been corrected.

7. PLOS authors have the option to publish the peer review history of their article (what does this mean?). If published, this will include your full peer review and any attached files.

Reviewer #1: No

Reviewer #2: **Yes: **Ahmed Idris Hassen

---

## [Editor Report · Acceptance letter]

1 Sep 2023

PONE-D-23-15692R1 

Legume-nodulating rhizobia are widespread in soils and plants across the island of O‘ahu, Hawai‘i 

Dear Dr. Nguyen:

I'm pleased to inform you that your manuscript has been deemed suitable for publication in PLOS ONE. Congratulations! Your manuscript is now with our production department. 

Kind regards, 

on behalf of

Dr. Ying Ma 

Academic Editor

PLOS ONE